# From Low-Grade Inflammation in Osteoarthritis to Neuropsychiatric Sequelae: A Narrative Review

**DOI:** 10.3390/ijms232416031

**Published:** 2022-12-16

**Authors:** Vladimirs Naumovs, Valērija Groma, Jānis Mednieks

**Affiliations:** 1Department of Doctoral Studies, Riga Stradins University, Dzirciema Street 16, LV-1007 Riga, Latvia; 2Institute of Anatomy and Anthropology, Riga Stradins University, Dzirciema Street 16, LV-1007 Riga, Latvia; 3Department of Neurology and Neurosurgery, Riga Stradins University, Dzirciema Street 16, LV-1007 Riga, Latvia

**Keywords:** osteoarthritis, low-grade inflammation, cytokines, signaling, central nervous system, neuropsychiatric disease

## Abstract

Nowadays, osteoarthritis (OA), a common, multifactorial musculoskeletal disease, is considered to have a low-grade inflammatory pathogenetic component. Lately, neuropsychiatric sequelae of the disease have gained recognition. However, a link between the peripheral inflammatory process of OA and the development of neuropsychiatric pathology is not completely understood. In this review, we provide a narrative that explores the development of neuropsychiatric disease in the presence of chronic peripheral low-grade inflammation with a focus on its signaling to the brain. We describe the development of a pro-inflammatory environment in the OA-affected joint. We discuss inflammation-signaling pathways that link the affected joint to the central nervous system, mainly using primary sensory afferents and blood circulation via circumventricular organs and cerebral endothelium. The review describes molecular and cellular changes in the brain, recognized in the presence of chronic peripheral inflammation. In addition, changes in the volume of gray matter and alterations of connectivity important for the assessment of the efficacy of treatment in OA are discussed in the given review. Finally, the narrative considers the importance of the use of neuropsychiatric diagnostic tools for a disease with an inflammatory component in the clinical setting.

## 1. Introduction

The existence of putative bidirectional communication between the brain and organ systems, including the musculoskeletal, has been proposed [1,2,3,4]. Virtually every somatic disease presents with inflammation at some point in its pathogenesis. There is a clear trend of coupling diseases with a strong inflammatory component with behavioral changes and mental disorders [5,6,7]. The crosstalk between peripheral inflammation and the Central Nervous System (CNS) is evidenced in neurological and psychiatric sequelae associated with chronic inflammatory joint diseases like rheumatoid arthritis (RA) [8]. The neurological manifestations of RA are usually uncommon, but the burden of cognitive impairment in RA can be significant [9,10,11]. Currently, the mechanisms that are utilized by the body to instantiate changes in behavior are not completely understood. There is also no clear-cut threshold of inflammatory process intensity beyond which it becomes pathological, behavioral changes occur, and a neuropsychiatric disease develops. Though the link is most evident in highly inflammatory and autoimmune diseases such as systemic lupus erythematosus, RA, and ulcerative colitis, evidence is substantially lacking in cases presented with low-grade chronic inflammation, not in the least because the hypothetical influence is temporarily spread out [12,13,14,15].

Osteoarthritis (OA) is a degenerative disease that affects the entire joint. Ageing, obesity, C and D vitamin deficiency, and traumatic injury to the joints are the primary risk factors [16,17]. Pain that comes in various variants: sharp, dull, predictable and unpredictable, acute and chronic, triggered by activity, constant, and nighttime, as well as stiffness and limitation of activities, are the three most prevalent symptoms of OA, reported by more than half of all patients [18,19,20,21]. Due to the aforementioned, OA is among the leading diseases in socio-economic burden [22,23]. Its prevalence and burden are projected to increase rapidly with an increasingly aging society.

Joint damage in OA occurs through repeated and excessive mechanical loading on the joint and the cumulative impact of low-grade inflammation over time or through injury sustained during the life course [24,25]. In the past, the disease has been viewed as degenerative, and loss of articular cartilage structure and function is one of the major hallmarks of OA [26]. This structural and functional alteration is characterized by a proinflammatory/catabolic state of the chondrocytes in the articular cartilage [27]. Pertinent literature suggests the existence of chondrocytes with inflammation-promoting and inflammation-diminishing features [28]. Currently, the contribution of chronic, low-grade inflammation (Figure 1) to the development of OA and the action of inflamed synovium as a trigger of the OA process has been suggested [25,29,30,31]. The synovial intimal cells are essential producers of various pro-inflammatory cytokines that drive the OA process in the affected joint [32,33,34,35]. Furthermore, these inflammatory mediators may activate their corresponding receptors on different brain cells and initiate various pathways of molecular signaling in the CNS [36].

In this review, we address OA as a condition that presents with chronic, low-grade inflammation and discuss the cross-talk between peripheral proinflammatory signals and the CNS.

### 1.1. Current Understanding of the Causes and Development of Osteoarthritis

Nowadays, OA is recognized as a common, multifactorial musculoskeletal disease [37,38]. Cell senescence, metabolic dysfunction, inflammation, and traumatic injury have all been proposed to play a role in developing the disease [39,40,41,42]. The hallmark of the disease is the existence of an imbalance between the destruction and formation of articular cartilage [43,44,45]. Due to changes in its molecular integrity, it is thought to begin with the structural decomposition of cartilage and subsequent increased susceptibility to trauma [46]. Trauma to the cartilage causes a release of damage-associated molecular patterns (DAMPs) into the synovial fluid and triggers an innate immune response from synovial cells. [47] The synovial cells produce inflammatory cytokines and DAMPs, thus recruiting other cells of the entire joint and causing a pro-inflammatory response [48]. Chondrocytes become hypertrophic and generate extracellular matrix-degrading enzymes such as matrix metalloproteinases (MMPs) and pro-inflammatory mediators in an attempt to reverse the damage to the cartilage (Figure 2), thus tipping the homeostatic balance to the side of catabolism [17]. Gradually, chondrocytes undergo structural and secretory pattern changes, as well as apoptosis [49,50]. Thereafter, cavities in degrading cartilage and subchondral bone created by dying apoptotic cells are invaded by blood vessels, thus establishing an avenue for inflammatory cells [51]. Immune cells release cytotoxic molecules that, together with the extracellular-matrix degrading enzymes, lead to the damage of joint tissues, subsequently causing a release of DAMPs, thus closing a feedback loop of the vicious circle (Figure 3) [52,53].

At this stage, the joint becomes a factory for producing low levels of proinflammatory molecules. These molecules then exert their action locally and at a distance. Due to the time over which OA develops, changes from persistent pro-inflammatory signaling may accumulate. As we shall see going forward, these accumulating effects include neuroplastic changes in different peripheral and central nervous system regions and ultimately lead to the alteration of behavior. 

### 1.2. Osteoarthritis and Mood Disorders

Accumulating evidence suggests that mood changes are reported by about one in ten OA patients. However, these are less prevalent than pain [17]. About one in five patients with OA are diagnosed with depression [54]. Greater pain scores, joint dysfunction, stiffness, and two or more comorbidities are associated with a higher incidence of depression among OA patients [55]. OA is associated with an OR of 1.27 for suicidal ideation and 2.09 for suicide attempts [56]. One patient among five OA patients will be diagnosed with anxiety disorder [57]. A greater risk of developing OA-associated dementia has been shown. However, no causality has been proven [58]. There is some evidence that senescence-associated processes and myopathy may influence the development of both OA and dementia [59,60,61].

Nevertheless, it is evident that there is a tendency for the coupling of the pathologic processes of OA and neuropsychiatric disease. A higher prevalence of mood disorders in OA patients that experience more pain, suggests chronic pain as the mediating mechanism of pathogenesis. Indeed, chronic pain itself has been shown to promote higher risks of developing depression, anxiety, and dementia [62,63]. On the other hand, chronic pain related conditions are known to worsen in patients with depressive symptoms, with patients showing lower pain thresholds [64]. Finally, it is highly likely that both conditions, mood disorders and chronic pain develop simultaneously and exacerbate one another.

## 2. Body Responses to Inflammation: Tissue Mediators and Signaling to the Brain

It should be acknowledged that the nervous and immune systems, directly or indirectly, play a regulatory role in virtually everything happening in the body. A temporary rise in body temperature, called fever, is an overall response from the body’s immune system. The body’s core temperature and, thus, fever mechanisms are coordinated by the thermosensitive neurons of the hypothalamus and, hence, are a prime example of peripherally-derived inflammatory cytokines affecting neuronal signaling [65]. However, it was unclear for a long time how peripheral inflammation leads to the changes in neuron firing in the hypothalamus. Throughout the 20th century, cytokines have been found to play an essential role in the modulation of inflammation and fever [66]. The injection of leukocytic pyrogen, later recognized as interleukin-1 (IL-1), into the preoptic anterior hypothalamus but not the other areas of the brain has been shown to cause the fever response [67]. Later, the significance of other tissue mediators, prostaglandins, was explored. The prostaglandin E (PGE) family was described together with antipyretic drugs, such as acetylsalicylic acid [68,69]. With further research, perivascular and cerebral endothelial cells were shown to possess the ability to bind IL-1 and produce PGE2 [70]. In turn, the latter was demonstrated to bind to the thermosensitive neurons and lead to fever [71]. Gradually, multiple pathways with different stimulus-response times gained their ground in explaining fever initiation [72,73,74]. One of the mechanisms described involves circumventricular organs (CVO). These are particular sites of the brain that lack the classical blood–brain barrier (BBB), and are thought to function as surveillance sites of the blood microenvironment by the CNS. The hallmark of CVOs is the presence of fenestrated endothelium and, as a result, increased permeability. Cytokines and pathogen-associated molecular patterns (PAMPs) are known to enter the brain and initiate sickness responses in proximity to the CVOs [75,76]. Another pathway involves the typical paracrine action of prostaglandins at the site of inflammation combined with the delivery to the CNS by crossing the BBB and using prostaglandin transporters of the plasma membrane of the cerebral endothelium [77,78]. 

Sensory nerves have also been implicated in signaling peripheral inflammation to the CNS. This pathway is thought to initiate non-specific responses to inflammatory stimuli, such as sickness behavior, and fever, as well as anti- and pro-inflammatory reflexes. Among the body functions regulated by the parasympathetic nervous system, the vagus nerve control of immune response should be mentioned [79,80,81,82,83]. However, it is up for debate whether vagal afferents play a significant role in fever production *per se*; it is nonetheless evident, that its terminals can sense inflammatory stimuli [84]. The vagus nerve terminals possess IL-1 receptors, whereas nodose ganglion cells display cytokine, toll-like receptors (TLRs), and PGE receptors [85,86,87]. It has been shown that PAMPs and cytokines can activate Kupffer cells in the liver, leading to the production of PGE2 and resulting in signaling inflammation to the CNS upon binding to the vagal terminals [88]. 

To recap the main points, inflammation may be signaled to the CNS using several pathways: (a) signal transduction through cerebral endothelium, (b) transport of signaling molecules across the fenestrated endothelium of the CVOs, (c) entry of signaling molecules into the brain via the BBB, and (d) ligation of the inflammatory signaling molecules to sensory nerve terminals and nerve signal transduction (Figure 4). These signaling pathways constitute gates for molecules, signaling peripheral inflammation, to enter the CNS. The fever case also demonstrates the physiological response mechanism of the brain to such stimuli. Overall, it implies that inflammation is listened to in different ways throughout the body by the nervous system as well, as shown readiness to respond to it.

### 2.1. Pro-Inflammatory Mediators: Signaling in Osteoarthritis Patients

Accumulating evidence suggests that there are prerequisites for cytokine signaling established in OA patients. Elevated levels of an array of pro-inflammatory cytokines, including IL-1, IL-6, TNF-α, chemokines, and PGE2 have been found in the serum of OA patients [89,90,91,92,93,94]. Similarly, plasma levels of PGE2 and 15(S)-hydroxyeicosatetraenoic acid (15-HETE), products of arachidonic acid metabolism are shown to be elevated in patients with symptomatic knee OA [94]. High levels of IL-1 have been found in the cartilage, synovial fluid, and synovial membrane of patients with OA [95]. The cytokine is also considered a major factor linking OA and obesity, one of the primary risk factors for disease development [96]. In rodent models of RA and OA, IL-1β has been demonstrated to play a significant role in the development of mechanical allodynia. The cytokine has been shown to act through dorsal root ganglion (DRG) neurons that possess IL-1 receptors. It has also been established that humans show expression of these same receptors in the neurons of DRG [97]. Patients with OA are also known to experience mechanical allodynia [98]. Furthermore, a subgroup of patients with OA was demonstrated to have increased expression of the IL-1β gene in peripheral blood leukocytes and higher pain scores [99]. Importantly, data about inflammation driven by DAMPs released from cartilage is based on proven evidence. However, there is no evidence suggesting that DAMPs can spill over into the circulation and exert their pro-inflammatory stimulation systemically [100]. Finally, different peripheral blood leukocyte compositions were recognized in elderly OA patients when compared to controls. The aforementioned results might signal the dysfunction of the immune system but merit further investigation [101]. Clinically, the link between OA and cardiovascular, as well as cerebrovascular events suggests the existence of systemic influence of local joint inflammation in OA [102]. Overall, there is enough evidence to consider that inflammatory signaling molecules, such as cytokines and arachidonic acid metabolism products, may leave the local site of inflammation in patients with OA. These signaling molecules were shown to exert their influence systemically in acute experimental inflammation. Though evidence of similar actions of these molecules is lacking in the case of OA, consider, for example, lack of fever as a prevalent symptom of OA. Nevertheless, links to other comorbidities, such as cerebral endothelial dysfunction, suggest systemic action of the aforementioned inflammatory molecules [102].

### 2.2. Mechanisms of Pain in Osteoarthritis

Neuropathic pain, and pain in principle, as we will discuss, are associated with peripheral inflammation. Furthermore, inflammatory mediators contribute to the processes of peripheral and central sensitization. Thus, mechanisms of transduction of noxious stimuli merit discussion in the context of crosstalk between peripheral inflammation and the brain.

Pain is the hallmark symptom of OA. The pain associated with OA is typically divided into two components: consistent, low, intermittent, unpredictable, and intense. It has been shown that unpredictable, intense pain episodes lead patients to refrain from physical and social activity [103,104]. It is also widely recognized that structural joint changes do not correlate well with symptom intensity in OA. Thus, other factors than joint deformity have been considered principal for pain development [105].

There are four types of neural organs contributing to the innervation of the knee [106]. Type one and two are corpuscular organs housing mechanoreceptors. Type three represent Aδ-myelinated fibers that respond to high threshold mechanical and thermal stimuli. Unmyelinated C-fibers form type four sensory organs. They are referred to as polymodal and respond to thermal, mechanical, and chemical stimuli [107]. Polymodal receptors are thought to play a major role in the pathogenesis of pain in OA [108]. These nerve fibers are believed to be silent under normal conditions and activated only when a pathological stimulus, such as inflammation in the case of OA, is present. Pain in OA likely begins as a nociceptive type in response to local inflammatory mediators. However, a neuropathic component develops as the disease progresses, ultimately leading to chronification [109]. There are two major components of chronic pain development in OA: (a) peripheral sensitization and (b) central sensitization.

#### 2.2.1. Peripheral Sensitization in Osteoarthritis

Peripheral sensitization describes a process by which DRG neurons become primed for excitatory stimuli. This happens by lowering the threshold needed to initiate an action potential. Thus, non-noxious stimuli may lead to incommensurable afferent signals, as well as the development of spontaneous signals. As described above, the inflammatory component plays a major role in the pathogenesis of OA. It is thought to be a leading contributor to the experience of pain as well. Inflammation-associated molecules, such as prostaglandins, bradykinin, cytokines, DAMPs, and nerve growth factor, are thought to ligate to sensory nerve fibers via transient receptor potential (TRP) channels, sodium channels, and mechanosensitive piezo ion channels [110]. This leads to different kinds of changes in neurons, including decreased receptor activation threshold and changes in receptor expression [111,112]. In addition to the inflammogenic sensitization, nociceptors may also express cytokines and chemokines, thus causing neurogenic inflammation and recruiting immune cells to the site of damage [113]. Accumulating evidence suggests that neuron-derived CCL2 regulates macrophage activation in the DRG after chemotherapy and leads to the development of neuropathic pain [114,115]. Additionally, the changes in sodium channel expression have also been shown to affect nerve signal transduction [116]. In summary, inflammation, both local and systemic, plays a crucial role in the development of peripheral sensitization [117]. This, in turn, enhances noxious stimulus transduction to the CNS, as well as causing reflective neurogenic inflammation. 

#### 2.2.2. Central sensitization, the Spinal Dorsal Horn Pain Processing System

The population of spinal dorsal horn neurons has long been the focus of neuropathic pain research and dates back to the establishment of the Gate Control theory by Melzack and Wall [118]. It has been postulated that noxious signals are inhibited via inhibitory interneurons, which themselves are stimulated by constant non-noxious signals. It is only when a noxious stimulus becomes greater than interneuron inhibition that balance tips and painful stimuli are allowed to reach higher CNS regions and lead to painful sensations. Hence, hallmark neuropathic symptoms, such as allodynia and hyperalgesia, as well as the phenomenon of chronic pain itself, have at least partially been attributed to pathologic changes of *the gate.*


The dynamic interactions between somatostatin-lineage excitatory interneurons and dynorphin-lineage inhibitory interneurons in the dorsal horn of the spinal cord have been determined as central to mechanical pain processing, as well to utilize mechanisms described in the Gate Control theory [119,120]. Other excitatory and inhibitory neurons have also been shown to play a role in neuropathic pain development; however, it remains beyond the scope of this review [121,122]. Nevertheless, it has been established that a disbalance of excitation and inhibition plays a determining role in the development of neuropathic pain and its chronification [123,124]. Various functional and morphological changes and their developmental mechanisms have been proposed and explored [125].

Microglial cells of the CNS were found to play an important modulatory role in spinal neuron synaptic plasticity in the chronification of pain [126,127,128]. Primary afferents were shown to produce colony-stimulating factor 1 (CSF-1) and to release it into the spinal dorsal horn when injured. CSF-1, in turn, activates microglial cells, which further release brain-derived neurotrophic factor (BDNF). BDNF then acts on excitatory interneurons in the dorsal horn to promote their excitatory drive [129]. The P2X7 receptor, which is primarily known in microglial cells for its immune signaling, when ligated by ATP, released from injured primary sensory afferents, leads to the release of pro-inflammatory cytokines and chemokines [130,131]. Its blockage has been shown to alleviate neuropathic pain [132]. Furthermore, P2X7 agonists were shown to reduce electroacupuncture-related neuropathic pain-alleviating effects, as well as enhance morphological changes in the neurons of the spinal dorsal horn [133].

#### 2.2.3. Central Sensitization, The Brainstem, Subcortex, and Cerebral Cortex

Descending regulatory pathways are also thought to play a crucial role in pain modulation. They are thought to be responsible for conveying emotional and cognitive state-specific effects, as many regions of origin of the descending pathways also participate in psycho-somatic affective sensations [134]. This modulatory control mainly originates from regions of the brainstem and produces its effects through the release of noradrenaline and serotonin. These pathways are also targets for tricyclic antidepressants and serotonin-noradrenalin reuptake inhibitors in the treatment of chronic pain [135]. Periaqueductal gray (PAG) matter is one such brain region. It has been implicated in autonomic responses to emotional states, and its disinhibition has been correlated to states of anxiety and depression [136,137]. PAG is known to exert pain-inhibitory modulation via the rostral ventromedial medulla (RVM) [138,139]. It is noteworthy, however, that RVM is known to convey both facilitatory and inhibitory pain-transduction signals to the spinal dorsal horn [140,141]. The locus coeruleus (LC) is another region that plays a part in pain modulation. It is a major source of noradrenaline in the CNS. Nociceptive pain input activates the LC, which in turn creates an attenuating effect on the dorsal horn pain-processing system, resulting in analgesia, and projects to higher cortical regions to produce hypervigilance and anxiety-like behaviors [142]. The LC is also thought to be the target region of gabapentinoids in the treatment of neuropathic pain [143]. It has been implicated in many psychiatric and neurologic conditions, such as Parkinson’s disease, post-traumatic stress disorder, anxiety, and insomnia, where arousal plays a seminal part [144,145,146,147]. 

Further pain processing occurs via dynamic brain region composition, collectively termed the pain matrix [148]. The pain matrix can be further subdivided into two components. The nociceptive network is represented by somatosensory systems and acts to pinpoint the type, time, and location of painful sensations. In turn, the salience network is responsible for the integration of painful sensations into episodic memory, emotional and cognitive states, and, ultimately, the initiation of top-down pain modulation. Unlike the nociceptive network, the salience network receives a wide array of different stimuli and is not limited to painful sensations, a feature that complicates the exploration of this particular network [149,150]. A full description of these networks is lengthy and remains out of the scope of this review. However, it is worth mentioning that spanning over the prefrontal cortex (PFC), anterior cingulate cortex (ACC), and anterior insular cortex (aIC), structural and functional changes of the salience network have been linked to such pathologies as chronic pain, depression, anxiety, mild cognitive impairment, dementia, and many others [151,152,153,154,155,156,157,158,159,160,161,162]. This, in turn, implies that the dysfunction of the process of salience attributed to bodily and environmental sensations is a major part of both chronic pain and mood disorders, potentially, linking these disorders. For instance, incorrect salience attribution to a painful stimulus might make it seem more painful. Similarly, objectively, minor personal failure might seem more tragic to a patient with general anxiety disorder. Clinically, correlations between chronic pain states and neuropsychiatric disorders are well recognized [163,164].

The mesolimbic dopamine (DA) pathway, thought to drive reward-seeking behavior, has been implicated in some comorbidities associated with chronic pain [165]. Furthermore, the contribution of microglia to the development of neuronal changes in the brain in the context of chronic pain has been proved. A blunted DA response in the context of chronic pain has been demonstrated after stimulation of the ventral tegmental area (VTA), a place of residence of the DA neurons of the mesolimbic pathway, with opioids. Further research has identified microglial BDNF signaling as a probable cause for this modulation [166]. Microglia and astrocytes were shown to influence top-down pain modulation in the PAG. Injection of astrocytic and microglial inhibitors in this region of the brain of rats with induced chronic constriction injury produced attenuated responses to noxious mechanical and thermal stimuli [167]. Similarly, microglial activation has been shown in the ACC [168]. Significant microglial activation has been found in the nucleus accumbens, VTA, and thalamus [169].

Central sensitization has long been recognized in a subset of OA patients. Hence, the usual comorbidities seen in patients with chronic pain, as well as treatment options targeting central sensitization, have been evaluated to some extent in patients with OA [170,171,172,173]. 

The PAG has been a region of interest for the treatment of chronic pain in OA. fMRI studies show altered functional connectivity between cortical regions, thalamus, and PAG in OA [174]. Furthermore, transcranial direct current stimulation of the PAG in the rat model of OA has shown some pain-alleviating capabilities [175,176]. Another study that was focused on electroacupuncture suggests that it might exert its anti-noxious effects by potentiating the endocannabinoid-mediated system at the level of PAG [177]. As indirect evidence for LC involvement in chronic pain of OA, GABA derivates may be used effectively to alleviate pain in OA [178]. However, this line of evidence is very limited. Changes in higher cortical areas are also described in OA patients. Reduced gray matter volume (GMV) in the ACC has been observed in patients with chronic pain syndromes [179]. Changes to the functional connectivity of the aIC and other pain-related regions, as well as disruptions in network connectivity, have been reported in OA patients [180,181]. However, overall empirical evidence is very limited.

Microglia involvement in the development of OA-associated chronic pain has been considered and explored [182]. In rodent models of OA, the administration of tocilizumab was shown to markedly reduce microglial activation in the spinal dorsal horn [183]. Laser moxibustion, a type of laser-assisted acupuncture, has been shown to effectively reduce mechanical hyperalgesia and decrease pro-inflammatory cytokine expression in the spinal horn of rodent OA models [184].

In summary, microglia play a role in the pathogenesis of neuropathic pain. They have been shown to modulate neuronal activation, thus influencing pain processing at every level. The exact mechanisms of this stimulation are not yet understood. However, many signaling pathways have been established. The fact that this process occurs at every level of pain processing raises the question of whether this phenomenon is pain-specific or whether abnormal signal processing by using other modality-specific systems might lead to similar pathology. Nevertheless, it becomes clear that a hallmark symptom of OA is either a driver of or, in part, a consequence of neuroinflammation. Moreover, the brain regions implicated in the development of chronic pain, are known to be affected in psychiatric conditions, which are known comorbidities to chronic pain.

## 3. Neuroinflammation 

Neuroinflammation is a complex response that recruits a myriad of cells. It has been reviewed in the context of a wide variety of neurological and psychiatric disorders [185,186,187,188]. The main contributors to the development of neuroinflammation are microglial cells and astrocytes [189]. Accumulating evidence suggests that the aforementioned cells are strongly associated with the development of neurodegenerative and psychiatric diseases [190,191,192]. Nevertheless, their exact roles in these diseases are still unclear. The role of inflammation is to provide host defense against exogenous pathogens and facilitate tissue homeostasis. Defense against pathogens includes the production of cytotoxic compounds, chemokines, and cytokines. The role of inflammation in homeostasis is the removal of tissue debris, promotion of damaged cell death, and regulation of tissue regeneration.

### 3.1. Microglia

Both neuroprotective and neurodegenerative effects of inflammation have been recognized in the nervous tissue [193,194,195]. Thus, neuroinflammation is heterogeneous in its presentation and depends upon *(a) temporal length, (b) degree, and (c) type of inflammatory stimulus.* Microglia are the cornerstone of neuroinflammation, responsible for surveying the brain environment, synaptic pruning, and initiation of host defense mechanisms [196,197,198]. The performance of this diverse milieu of tasks requires functional plasticity. Conventionally, microglial cells are divided into M1 and M2 phenotypes based on dynamics of activation and arrays of secreted molecules. M1 is considered to be “classically activated” and thus pro-inflammatory, whereas M2 is “alternatively activated” and represents a pro-regenerative phenotype [199]. Nevertheless, it should be noted that this dichotomous differentiation is merely ease of use. Microglial activation, secretion patterns, and functions are complex and should be viewed as a spectrum rather than a bipolarity. 

Ramified microglial cells are distributed throughout the brain in a net-like manner. This particular distribution is tightly regulated [200]. CNS injury leads to the selective proliferation of pro-inflammatory microglia at the site of injury [200,201]. When the issue is resolved, spatial distribution and numbers of microglia are reestablished via apoptosis and egress of cells [202]. This restored microglial pattern may be distinct in its population functional activation status, for instance, by acquiring a pro-regenerative state in the majority of microglial cells rather than the ramified, surveilling type [203]. The ramified microglial cells constantly survey the microenvironment, looking for molecular patterns associated with pathogens (PAMPs) and patterns associated with cell damage (DAMPs) [204,205]. In doing so, the cells utilize Toll-like receptors (TLR) and nuclear oligomerization domain-like receptors [206,207,208]. It has been shown that microglial cells abundantly express TLR4 on the surface of their processes. This has also been implicated in playing a role in the development of chronic inflammation in some neurodegenerative diseases [209,210]. It has also been shown that inhibition of the TLR4/NF-kB pathway promotes the switching of the microglial cell from the pro-inflammatory to the pro-regenerative phenotype [211,212].

Changes in secretion patterns and morphology of the microglial cell occur when it encounters molecular patterns it recognizes. These changes depend on the degree of insult, molecular signaling array, and the previous state of the cell [213,214]. In the case of *“classical activation”*, a microglial cell starts to secrete Il-1β, TNF-α, IL-6, nitric oxide, reactive oxygen species, and proteases that serve to damage the pathogen and prepare it for phagocytosis. However, simultaneously, these molecules can also cause damage to the host cells in the vicinity [214,215]. Microglia can also be activated via IL-4, IL-10, IL-13, and TGF-β. This activation pattern is termed *“alternative activation”* and is associated with the secretion of neuroprotective molecules, such as IL-10, Chi3l3, FIZZ1, Arginase 1, Ym1, CD206, IGF-1, and Fzd1. This molecular milieu is associated with the clearance of tissue debris and repair of extracellular matrix, functions essential for regeneration [214,215]. Thus, it is crucial that the switching of the microglial activation pattern occurs unrestricted at a precise moment. Failure to switch from the M1 to M2 phenotype may result in protracted autoimmune damage to the CNS, creating an even more pro-inflammatory environment. This may occur in the presence of ongoing tissue damage, production of pro-inflammatory cytokines, or disruptions in microglial intracellular machinery. 

### 3.2. Astrocytes

Astrocytes are the most numerous glial cells in the CNS. Each astrocytic cell has its designated area of influence that does not overlap with other astrocyte areas. These cells form networks via tight junctions. Astrocytes contact virtually every type of cell in the CNS. They participate in the formation of the BBB, regulate neuronal metabolism, serve as an ionic and neurotransmitter buffer, are involved in synapse formation and pruning, as well as participating in synaptic information processing [216,217,218,219,220,221,222]. Not unlike microglia, astrocytes are a heterogeneous group of cells [223]. Structural and receptor expression differences contribute to the functional specialization of each cell, as well as their functional diversity as a population of cells. This heterogeneity is evident even in a small brain region [224]. When challenged with tissue damage, astrocytes contribute to what is termed “reactive astrogliosis”. However, this process can lead to both neuroprotective and neurotoxic resolutions [225]. Recently, astrocytes were subdivided into two functional populations termed “A1” and “A2”, analogous to the microglial M1 and M2 [226]. A1 is thought to be a neurotoxic subtype. The proliferation of this subtype after experimental, non-lethal ischemic stroke was associated with greater tissue damage [227]. On the contrary, A2 is the neurotrophic subtype. A2-induced proliferation after ischemic injury was associated with maintained tissue morphology [227]. The existence of intimate bidirectional communication between astrocytes and microglia was shown. Microglia regulate astrocytic functions via the secretion of cytokines and growth factors. Microglia-derived TNF-α and IL-1α promote astrocytic differentiation into the A1 subtype. Astrocytes, on the other hand, can modulate microglial functions by secreting calcium ions, cytokines, chemokines, and other inflammatory mediators [228]. 

## 4. Emerging Imaging Technologies Used for the Detection of CNS Alterations in Osteoarthritis

The CNS alteration in OA is likely to result from some secondary neuroinflammatory changes, therefore, there is a genuine need for anatomical and functional CNS studies to explore the brain structures involved, as well as functional brain networks and areas in OA. Until now, the structural and functional alterations of the brain that appear due to inflammatory changes are more extensively studied in RA than in OA patients. In a longitudinal study of RA, Scherpf et al. (2018) explored how chronic peripheral inflammation affects brain connectivity and morphology. They showed a reduction in gray matter volume in the left lingual gyrus and left inferior parietal lobe. These findings were correlated to an increased ESR, one of the markers for active inflammation [229]. Furthermore, MRI analyses evidenced connectivity alterations in a wide range of neuronal networks, including default mode network (DMN), salience network (SN), and others. Studying the relationship between major depressive disorder with peripheral low-grade inflammation, Opel et al. (2019) found a correlation between the levels of CRP and gray matter volume reduction in PFC, insula, and the temporal cortex [230]. In the study of Lan et al. (2020), neuropsychological assessment and fMRI were performed before and a week after OA knee arthroplasty to determine changes in cognitive ability reflected by the assessment scores, functional connectivity, and regional brain activity [231]. Patients with knee OA showed a significantly lower amplitude of low-frequency fluctuation, correlating with spontaneous neuronal activity (ALFF) in bilateral angular, medial superior frontal gyrus, and precuneus. Another study showed a correlation between systemic inflammation, as judged by hs-CRP levels, and reduced functional connectivity within the frontotemporal network (FPN). Reduced FPN connectivity was positively correlated to the number of subjects reported as “discussion partners”, and there was an association between systemic inflammation and the size of the subject’s social-network [232]. Yet another study found reduced activity in the anterior insula, amygdala, hippocampus, and temporal lobe in response to positive *versus* neutral images, which correlated to increased peripheral levels of CRP and IL-6 [233]. It has been demonstrated that hip replacement surgery can reverse left thalamus volume loss in OA patients [234]. Therefore, longitudinal studies of the gray matter volume changes and connectivity alterations have become increasingly important in the assessment of treatment efficacy in OA. Indeed, the studies focused on brain volume alteration may be applied to various populations, and the existing studies can be reproduced with the high availability of advanced MRI analysis freeware tools such as SPM (Wellcome Department of Imaging Neuroscience at University College London) or Freesurfer (Laboratory for Computational Neuroimaging Athinoula A. Martinos Center for Biomedical Imaging). Apart from voxel-based morphometry, which is the most widely used method of brain volumetry, surface-based volumetry studies are becoming increasingly popular. Similar to volumetry studies, a wide range of functional MRI analysis methods are currently available in various software packages, e.g., general linear model (GLM) and independent component analysis (ICA). So, whenever a comparison between various studies and MRI methods is carried out, the use of MRI data for a particular OA patient should be kept in mind, offering a promising option in the treatment and follow-up by monitoring the functional network architecture of the brain [235,236]. The localization of OA-affected joints, as well as the side of disease-caused damage, have been shown as important factors that correlate with regional brain volume changes. It has been shown that knee OA affects cortical gray matter volume differently to hip joint damage [237]. Importantly, OA induced pain, rather than the involvement of a particular joint, could also determine brain volume changes, since the brain pain matrix is unambiguously affected in all symptomatic cases (Figure 5). Both OA and fibromyalgia patients have been shown to display regional brain volume changes caused by pain [238]. As discussed above, there is ample evidence of functional connectivity changes in some brain regions in the context of chronic pain [239,240,241]. Effective non-pharmacological therapies, such as cognitive-behavioral, acceptance, and commitment therapy for chronic pain, were shown to alter functional connectivity and neural activation dynamics in the salience network (SN), default mode network (DMN), and FPN. These changes of functions across the SN, DMN, and FPN was then associated with positive outcomes [242].

## 5. Neuropsychiatric Phenomena in the Context of Chronic Peripheral Inflammation

A link between peripheral inflammation and neurocognitive symptoms has been known for a long time. Over the past 30 years, a well-known constellation of symptoms for anyone that has ever suffered from infectious disease has become collectively known as “sickness behavior”. Fatigue, reduced locomotion, somnolence or sleep disturbances, and anorexia are the hallmark symptoms of acute infection [243]. IL-1β has been deemed the principal inductor of sickness behavior in the CNS [244].

First, it was shown, that administration of LPS resulted in non-rapid eye movement sleep increase, hyperalgesia, and conditioned aversion [245,246]. It was then further shown that LPS induces a decrease in the rapidness of locomotion [247]. Interesting results were achieved concerning motivation. Healthy volunteers after the LPS challenge were more likely to act upon low-effort-low-reward options and less likely to follow high-effort-high-reward options [248]. Memory function has also been shown to be decreased after LPS injection [249]. In vivo changes in cortical neuron excitation in frontal and motor cortices in response to LPS injection have been described by Odoj et al. (2021). They observed an increase in spontaneous Ca^2+^ transients 5 h after challenge with LPS in layer 2/3 cortical neurons and suggested the hyperexcitability is likely driven by layer 5 pyramidal cells but not the somatosensory cortex neurons. Decreased excitability of GABAergic interneurons was also observed. They further showed the effect’s dependence on preserved TNF-α signaling pathways and their independence from the presence of microglia [250]. Overall, these results show that peripheral inflammation indeed causes direct changes in neuronal signaling. However, surprisingly, at least in the case of acute inflammation, these changes do not occur due to microglia-derived pro-inflammatory factors. It is also important to note that inflammation tips cortical excitation/inhibition balance to the excitation side. 

Inflammation has been shown to alter glutamate signaling and metabolism, ultimately leading to increased extrasynaptic availability of glutamate. Astrocytes play an important role in glutamate metabolism. They capture glutamate in the synapses via excitatory amino acid transporter (EAATs) isotopes 1 and 2, carry out endocytosis, and deactivate glutamate to glutamine. Then glutamine is released, and neighboring neurons can reuptake it [251]. Activated microglia has been shown to secrete pro-inflammatory cytokines, including TNF, IL-1β, and IL-6. Decreased expression of glutamate transport has also been demonstrated [252,253]. Microglial cells express glutamatergic AMPA and NMDA receptors, whereas, after pro-inflammatory activation, microglia have been shown to also express EAAT-1 and 2. When these receptors are ligated by glutamate, an increase in the release of cytokines occurs [254]. Excess glutamate may bind to NMDA receptors on neurons, resulting in hyperexcitability and, as a consequence, incorrect signal processing or excitotoxicity [255]. Hence, glutamate dysregulation has been implicated in several pathologies (Figure 6) [217,256,257]. 

Dopamine dysregulation has been implicated in depressive patients with increased proinflammatory markers. These patients show a diminished response to SSRI therapy but respond well to antidepressants with dopaminergic action [258]. This data complements the literature on the effect of peripheral inflammation on the mesolimbic reward circuitry [259,260]. Dopaminergic neurons projecting from VTA to the ventral striatum and ventro-medial prefrontal cortex are thought to be responsible for motivation and reward. Reduced dopaminergic signaling in this pathway has been associated with anhedonia, a principal symptom of both depression and sickness behavior. Hence, a pharmacologic increase in dopamine signaling may alleviate the symptoms of anhedonia [261]. 

Sickness behavior is associated mostly with acute inflammation in the presence of infection. However, the same cytokines thought to be responsible for this behavioral pattern are present and exert systemic actions in other inflammatory diseases, such as RA, systemic lupus erythematosus (SLE), and OA [96,262,263]. Patients with RA and SLE are known to suffer from fatigue and depressed mood [264,265]. However, there is no clear correlation between disease activity and subjective feelings of fatigue [266]. Fatigue is also a common symptom of OA [267]. Overall data is lacking on sickness symptoms in OA and rheumatic diseases. However, there is substantial literature available concerning neuropsychiatric conditions and inflammatory diseases. RA has been associated with an increased risk of developing anxiety and major depressive disorder [268,269,270]. In SLE, neuropsychiatric symptoms have long been recognized in the American College of Rheumatology diagnostic guidelines [13]. These symptoms include conditions ranging from mood disorders to full-blown psychosis and cognitive dysfunction. OA has also been shown to pose a higher risk of developing mood disorders [271,272]. Furthermore, the association between OA and dementia has been shown [58]. NSAIDS were shown to be neuroprotective against dementia in the case of OA, suggesting inflammation as a link between the diseases [273]. However, there is some evidence that the OA-associated immunological phenotype might play a neuroprotective role [274]. An increased risk for the development of RA-associated dementia has also been reported [275,276]. Similarly, treatment of RA with biologics, namely the TNF inhibitor etanercept, showed a degree of neuroprotection compared to methotrexate [277]. TNF-α inhibitors were also studied for their potential in the treatment of dementia outside rheumatic diseases and OA, showing a degree of efficacy [278].

## 6. Discussion

Pertinent literature suggests that low-grade inflammation might play a central role in the development of OA and associated symptoms. Damaged cartilage releases DAMPs that stimulate synovial cells to produce proinflammatory cytokines in an attempt to jumpstart the innate immune response. Chondrocytes, stimulated by the inflammatory mediators, release proteinases that destroy the cartilage around them, thus releasing more DAMPs into the synovium. With time, blood vessels grow into the synovium and establish a gate for circulating immune cells. However, by that time, *circulus viciosus* is established and a joint becomes a local factory for proinflammatory molecules. 

Inflammatory mediators then gain the ability to be released into the bloodstream and exert their proinflammatory action at a distance from the inflamed joint. One such particular organ is the brain. Acting on the endothelial cells or diffusing into the brain parenchyma through the CVOs, proinflammatory cytokines stimulate astrocytes and microglia that, in turn, start to produce their inflammatory mediators. These inflammatory mediators influence neuronal activity, ultimately leading to the development of the usual catarrhal symptoms. These proinflammatory cytokines then travel and reach the brain. Ultimately, there are two ways for these molecules to reach the CNS: via circulation through cerebral blood vessels and via primary sensory afferents, and further through the spinal cord. On the receiving end, glial and neuronal cells exhibit the ability to sense these signals and change their function accordingly. Microglia and astrocytes start producing inflammatory cytokines of their own, and neurons change their firing patterns. Due to changes in firing patterns, the remodeling of neuronal circuits occurs, as evidenced by the body of literature on neuropathic pain. Literature is lacking on the exact types of changes that occur in the CNS in the case of inflammation. Endotoxemia studies show that there are acute behavioral effects, as well as the alteration of neuronal excitability after LPS injection. However, these do not explain the accumulating evidence of the connection between peripheral inflammation and neuropsychiatric sequelae, giving strength to the factor of time. Indeed, a major part of neuropsychiatric manifestations of peripheral inflammation is neurodegenerative diseases, as evidenced by connections of Alzheimer’s disease and Parkinson’s disease with inflammation [198,279,280,281]. Neurodegeneration does not occur quickly but rather develops over time, as evident by the age of onset of 65+ and the further slow progression of disability [282]. Nevertheless, fMRI studies show changes in functional networks, mainly SN, DMN, and VAN, in the context of systemic inflammation. These changes might explain some phenotypical phenomena, such as mood disorders and psychotic manifestations [283,284,285]. Finally, Goodkind et al. (2015) found an overarching pattern of reduced gray matter volume in the bilateral insula and dorsal-anterior cingulate cortex—roughly corresponding to the SN, as well as correlated these changes with poorer executive functioning [286]. 

Ultimately, the topic of immune-to-brain communication remains understudied. Especially when taking into account the polymorphism of inflammatory process manifestations in the body, complex, multivariate pathways of communication of the immune system with the brain, and a spectrum of changes in the CNS that might accommodate the neuro-immune communication. Further research is needed to elucidate the intricacies of communication, as well as to potentially establish new therapeutic targets for neurodegenerative and psychiatric diseases.

## 7. Conclusions

In summary, we show that chronic low-grade inflammation seems to be one of the essential intricate mechanisms implicated in the development of OA. We then discuss potential ways the inflammatory signal might reach the CNS. Further, we discuss changes that occur in the CNS in the context of peripheral inflammation and show their correlation to changes in behavior as well as neuropsychiatric disease development.

## Figures and Tables

**Figure 1 ijms-23-16031-f001:**
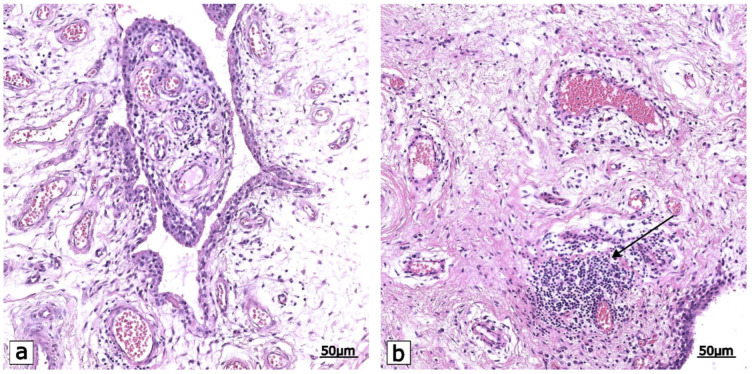
A representative image that depicts the histopathological features of low-grade synovitis in OA. (**a**) The synovial lining layer is slightly thickened, and the stromal density is slightly increased; few perivascular lymphocytes are evident. (**b**) The lining cells form 3–4 layers, the cellular density within synovial stroma is increased, and inflammatory infiltrate is present (arrow). Hematoxylin and eosin staining. Scale bars: 50 μm.

**Figure 2 ijms-23-16031-f002:**
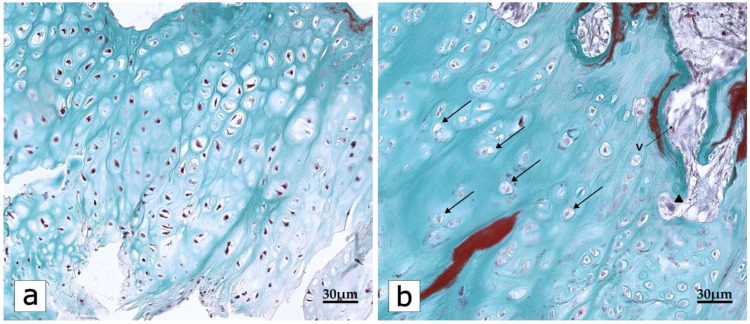
A representative image that depicts the OA-associated changes in Safranin O/Fast green stained section of articular cartilage. (**a**) Degradation of cartilage and changes in the proteoglycan content, as indicated by the lack of orange–red staining, is evident. (**b**) Enlarged, hypertrophic chondrocytes contain abundant cytoplasmic inclusions (arrows). Deep vascular invasions (V) into the cartilage matrix and reactive chondroclasts (arrowheads) are evident. Safranin O/Fast green staining. Scale bars: 30 μm.

**Figure 3 ijms-23-16031-f003:**
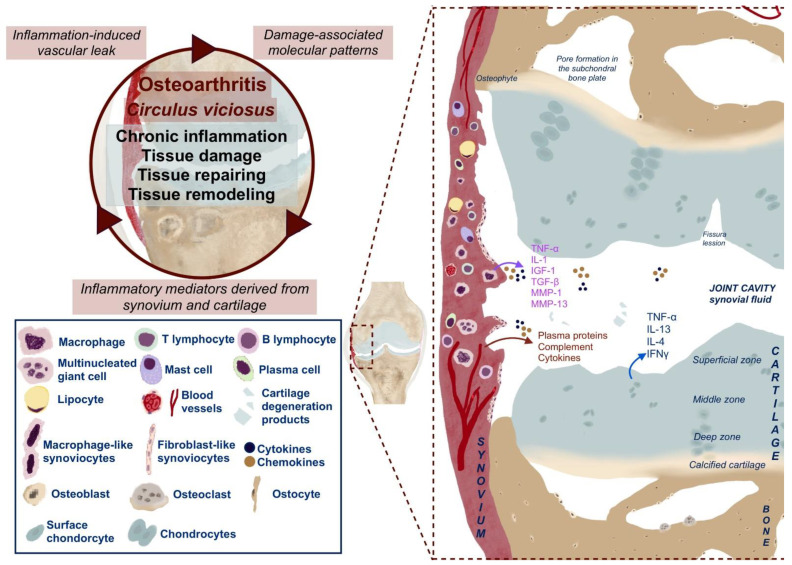
Signaling pathways and structural changes associated with the development of osteoarthritis.

**Figure 4 ijms-23-16031-f004:**
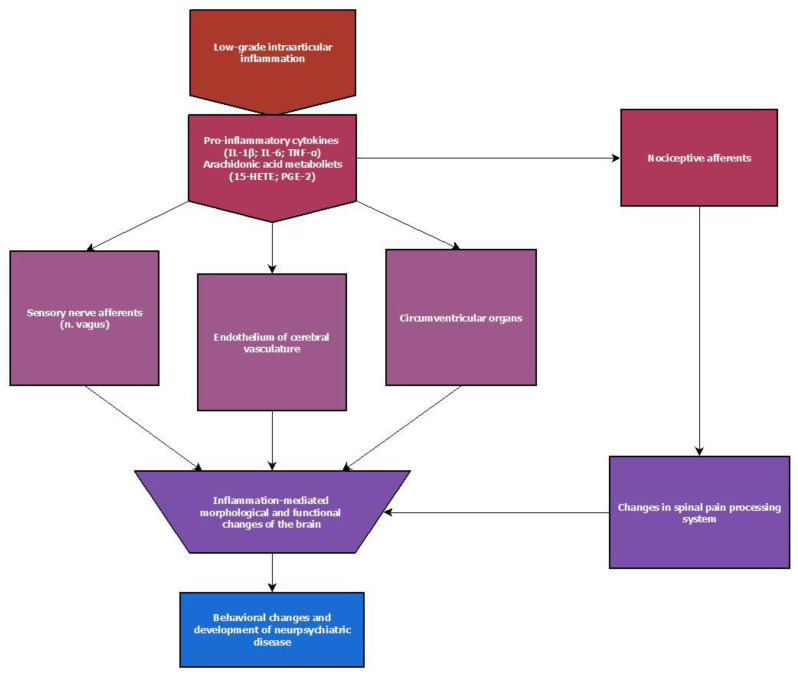
An outline of the main pathways for signaling inflammatory mediators to the central nervous system and the sequelae of recognized routes.

**Figure 5 ijms-23-16031-f005:**
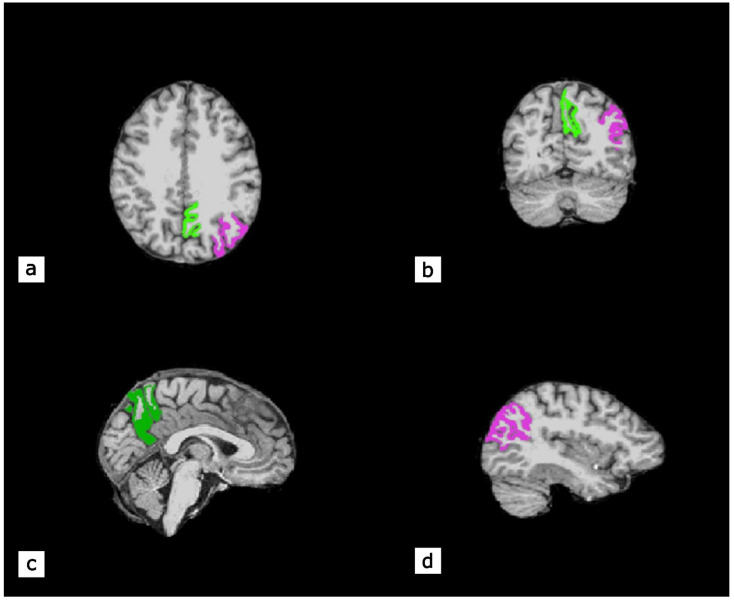
An illustrative example of magnetic resonance imaging-based brain volumetry analyses of functional neuroimaging data in a patient with OA confirms the alteration of the structures in the left parietal lobule. Surface-based volumetry demonstrates *precuneus* (colored in green) and inferior parietal gyrus (colored in purple). (**a**) Axial plane; (**b**) Coronary plane; (**c**) Medial sagittal plane; (**d**) Lateral sagittal plane.

**Figure 6 ijms-23-16031-f006:**
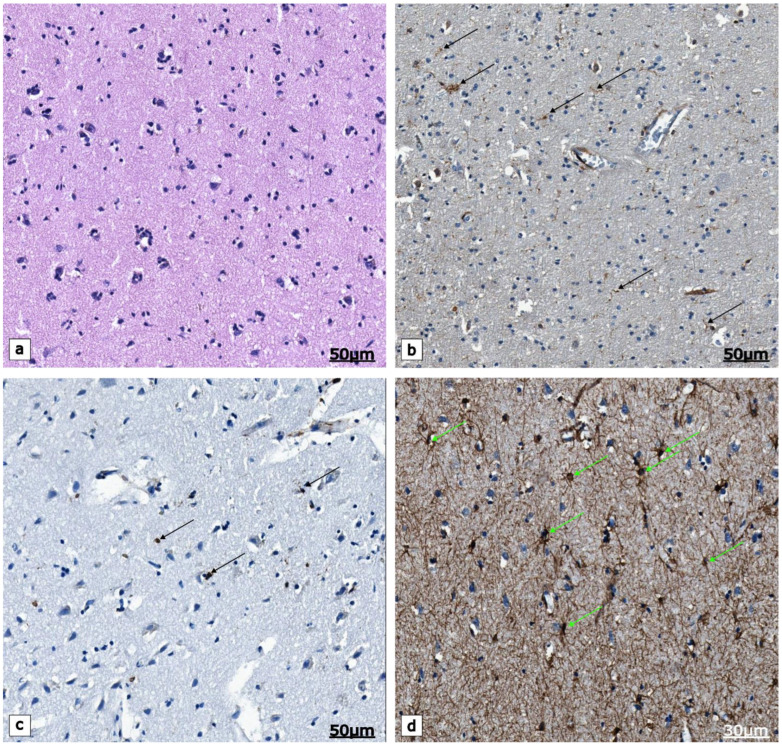
The structural appearance of the inferior parietal lobule of OA patient, post-mortem bran tissue. (**a**) An overview of the lobe as observed by the use of routine staining; (**b**) The interface of the gray and white matter presents with brown coloration in the cell bodies and moss-like processes of microglial cells (arrows), Iba1 immunohistochemistry; (**c**) Microglial cells (arrows) visualized by the presence of brown reaction products confirmed by the use of an anti-CD68 antibody, CD68 immunohistochemistry; (**d**) Astrocytes (arrows) that demonstrate strong expression of the glial fibrillary acidic protein (GFAP), GFAP immunohistochemistry. Scale bars: (**a**–**c**) 50 μm; (**d**) 30 μm.

## Data Availability

A publicly available bibliographic database, PubMed.gov, was used in this study. The full bibliographic reference list is available upon request from the corresponding author.

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
