# Peer review of "From Low-Grade Inflammation in Osteoarthritis to Neuropsychiatric Sequelae: A Narrative Review"

_ijms, 2022, doi:10.3390/ijms232416031_

Round 1
Reviewer 1 Report
Here, Vladimirs et al have made an effort to compile the development of neuropsychiatric disease in the presence of chronic peripheral low grade inflammation with a focus of brain signalling.
The review overwhelms the reader with a lot of information but is not opinionated. It doesn’t imply authors point of view or interpretation.
The review includes 281 citations which almost cites each sentence and the point to be conveyed is missed.
The figures added to the review are not clear and need to be changed.
Overall, the review doesn’t add interpretation to the current field but is a summary of all the study till date.
Author Response
The 1st comment: Here, Vladimirs et al have made an effort to compile the development of neuropsychiatric disease in the presence of chronic peripheral low grade inflammation with a focus of brain signalling.
Response: We would like to thank the reviewer for taking his/her time and giving useful comments on our research. Indeed our scientific paper aims to review the mechanisms by which the pathological process of OA may lead to the development of neuropsychiatric disease.
The 2nd comment: The review overwhelms the reader with a lot of information but is not opinionated. It doesn’t imply authors point of view or interpretation. The review includes 281 citations which almost cites each sentence and the point to be conveyed is missed.
Response: We thank the reviewer for his/her valuable comment. Concerning the overwhelming amount of information and citations, we believe that it is important to give the reader a comprehensive overview of the field. As the review strives to tie together the field of musculoskeletal disease, neurology, and psychiatry, a comprehensive overview of each of the mentioned fields is required to help the reader understand the main points. We further would like to direct the reviewer’s attention to the following lines in the manuscript that, we believe, serve to interpret overviewed literature: 213-220; 224-228; 388-389; 447-451; 508-522; 565-569, as well as “Discussion” section of the manuscript.
We also believe it is important to mention that all cited literature is interpreted by the authors, and citations not only convey the main findings of the cited articles but serve to build a comprehensive narrative.
Nevertheless, we do agree that the review would benefit from further clarifications; thus, we have added explanatory and summarizing remarks. Please address the following lines: 99-104; 127-134; 142-143; 180-184; 341-345; 391-393.
The 3rd comment: The figures added to the review are not clear and need to be changed.
Response: We thank the reviewer for the comments and detailed considerations. Our tailored Figures are diverse and, therefore, attractive to the reader; these include original drawings, schemes, and images of tissue samples. Secondly, following the MDPI requirements, all Figures are submitted at a sufficiently high resolution (minimum 1000 pixels width/height, or a resolution of 300 dpi or higher) and in a format recommended. Third, the images of the tissues of interest are unique since reflecting simultaneously both fields of interest ‒ the entire joint affected by OA and the neurological involvement of the CNS confirmed in the OA subject. The tissue images are captured with a Glissando Slide Scanner (Objective Imaging Ltd., Cambridge, UK). Reproducible images are obtained using the Aperio ImageScope program v12.2.2.5015, Leica Biosystems Imaging, Vista, CA, USA, and images are processed with the ImageJ program (National Institute of Health, Bethesda, MD, USA). Considering the aforementioned reasons, we disagree with the reviewer on the necessity to change the figures. Nevertheless, we do agree that better clarity can be achieved, thus have made changes to figures 1, 2, and 6 and their respective explanatory notes.
The 4th comment: Overall, the review doesn’t add interpretation to the current field but is a summary of all the study till date.
Response: We thank the reviewer for his/her comment. Our review aims to demonstrate, using pertinent literature, the case for neuropsychiatric sequelae following the OA pathological process, as well as to offer a narrative that would explain hypothetical mechanisms of such developments. Though low-grade inflammation has been implicated before as a driving factor in neuropsychiatric disease, extensive reviews in the context of OA are lacking. Thus, through reviews such as the one we humbly present, we hope to facilitate research development in the field. Furthermore, the topic of neuropsychiatric disease in the case of OA is not widely appreciated. In this light, a review making a strong case for neuropsychiatric comorbidities in OA might help shed light on the problem.
Reviewer 2 Report
Congratulations for your research.
I think the relationship between musculoskeletal injuries and disorders of the central nervous system is very interesting.
Perhaps it could be advisable to indicate on line 34 the meaning of CNS (Central Nervous System), even though it is a widely used abbreviation.
The good description of the histopathological and metabolic changes in OA is noteworthy. Futhermore, the pain control mechanisms information is attractive.
Would a study design be possible in which to analyze the correlation between different OA treatment options and changes in the lesion and the CNS? What changes could occur with physical exercise in OA?
Author Response
The 1st comment: Congratulations for your research.
I think the relationship between musculoskeletal injuries and disorders of the central nervous system is very interesting.
Perhaps it could be advisable to indicate on line 34 the meaning of CNS (Central Nervous System), even though it is a widely used abbreviation.
The good description of the histopathological and metabolic changes in OA is noteworthy. Futhermore, the pain control mechanisms information is attractive.
Response: We would like to thank the reviewer for taking his/her time and giving useful comments on our research. We also find it pleasing to see the interest in the topic. We have addressed the abbreviation concern and have added an explanatory remark on line 34 of the manuscript.
The 2nd comment: Would a study design be possible in which to analyze the correlation between different OA treatment options and changes in the lesion and the CNS?
Response: We thank the reviewer for this comment. We do believe it is important also to address the role of therapy concerning neuropsychiatric phenomena and their corresponding brain changes in the context of OA. A study like this would be of high clinical importance as well. However, significant limiting factors are present.
First of all, it is still not well understood what kind of brain changes occur in OA per se. Extensive research is needed to describe the brain involvement in OA and then to further correlate it to the clinical phenomena of the neuropsychiatric disease, being represented by mood disorders or even neurodegenerative disease. This requires animal-model studies, brain fMRI studies of OA patients, and post-mortem brain tissue studies of OA patients.
Furthermore, OA is a heterogeneous disease, both in its phenotype, in the presentation of symptoms, and endotype, as in the pathogenic mechanisms which lead to the development of the disease. Right now, there is an accumulating body of literature on the topic; however, no consensus has been reached. Stratification of patients by their main pathologic drivers is important in a study like this, as it will most probably influence baseline risks of developing neuropsychiatric sequelae. To offer an example, a patient with inflammation driven by senescent cells will likely have more circulating cytokines, hence higher risks of developing neuropsychiatric disorders, than patients whose OA developed solely in the context of metabolic dysregulation. For an in-depth review of the current literature on the topic, please refer to Mobasheri et al. 2019. https://pubmed.ncbi.nlm.nih.gov/30461544/
Another limiting factor relies on the evaluation of the therapeutic effect on brain changes. To this day, psychopharmacology relies on indirect measurements when assessing therapeutic outcomes for mood disorders. This usually is in the form of self-report questionnaires of patient-perceived symptoms of the disease in question rather than direct measurements of brain changes. However, changes in the scores of these questionnaires are not well correlated to the factual changes in brain circuits. This is because brain interrogation techniques are either highly invasive or economically burdening. With the development of functional magnetic resonance imaging techniques and our understanding of brain functional dynamics in the context of health and disease, this might become a relatively economically-sound and non-invasive option.
Ultimately, it is possible to design a study to assess different OA treatment option effect, pharmacologic or non-pharmacologic, on the risk of developing neuropsychiatric disease and the ability to alleviate their symptoms. It would be a much more complicated task to assess treatment option effects on the brain changes themselves and, frankly, probably too early of a task for the field.
The 3rd comment: What changes could occur with physical exercise in OA?
Response: We thank the reviewer for this question. Physical exercise indeed emerges as a significant therapeutic tool of intervention not only in diseases of the musculoskeletal and cardiovascular systems but also in the reduction of overall risks of developing age and lifestyle-related pathologies. Especially where a low-grade inflammatory component is present.
Moderate physical exercise has been a recommended therapeutic intervention for patients with OA. Pertinent literature suggests that it is as good, maybe even superior to conventional NSAIDs, in reducing pain sensation and facilitating limb function. For a comprehensive meta-analysis, please address Farsen et al. 2015. https://pubmed.ncbi.nlm.nih.gov/25569281/
Physical exercise has also been shown to produce positive effects on depressive and cognitive symptoms in patients suffering from neurodegenerative disease. See Dauwan et al. 2021 https://www.ncbi.nlm.nih.gov/pmc/articles/PMC7990819/
The literature is still lacking on the exact mechanisms physical exercise utilizes in its positive influence on cognition and mood. However, there is data showing that exercise can positively influence inflammatory processes in the body, namely, modulate levels of circulating cytokines. An indirect effect on inflammation is via the reduction of adipose tissue and, hence, the reduction of secreted adipokines. Furthermore, it is believed that the positive effect on mood disorders, at least in part, stems from promoting brain neurotrophic factor release and vascular proliferation. This constellation of effects is hypothesized to enhance neuroplasticity, thus positively affecting learning, memory, and cognition, ultimately serving a neuroprotective function. Another indirect effect of exercise on brain health is in sustaining the BBB. As leaky BBB is a direct consequence of neuroinflammation, which itself is both present in many neuropsychiatric disorders and, at least in part, a consequence of peripheral inflammation, reduction in the overall inflammatory background of the organism is beneficial to BBB integrity. For a more detailed description, please see Scheffer et al. 2020 https://www.ncbi.nlm.nih.gov/pmc/articles/PMC7188661/
It is important to note that physical exercise follows a hormetic effect. In a nutshell, hormesis refers to the dose-response concept of stress-repair mechanisms in biological organisms. It postulates that a certain amount of stress and even damage is beneficial to the organism as it further promotes adaptation and regeneration. Physical exercise has been shown to provide such stress in the form of oxidative stress, a by-product of any metabolic activity. However, too much stress might be detrimental and reduce the positive effects of exercise or even completely reverse them. It is still a matter of debate how much exercise is in the Goldilocks zone.
For a review of the concept, please refer to Calabrese EJ 2018. https://www.ncbi.nlm.nih.gov/pmc/articles/PMC7498668/ For a review of hormesis in physical exercise, please address Powers et al. 2020 https://www.ncbi.nlm.nih.gov/pmc/articles/PMC7498668/
In summary, physical exercise might provide a systemic beneficial effect both to the pathological mechanism of the OA and downstream brain changes. However, the exact amount of exercise is important, and overstraining may lead to detrimental effects.
Reviewer 3 Report
Dear Authors,
thank you very much for providing this extensive narrative review tying to link neuropsychiatric and inflammatory processes in osteoarthritis.
Due to the length of you paper I will only recommend some general issues:
Please ensure figures are cited correctly if you did not produce yourself.
L.175 Rewording recommended for: "Are also shown to suffer"
If I was able to follow your explanations correctly and based on the fact, that you cited hundreds of papers but some of the essential assumptions are more connected by theory, I would recommend to change in the beginning of your discussion to a less absolute conclusion, like "might play an important role".
Good luck with the paper.
Author Response
The 1st comment: Dear Authors, thank you very much for providing this extensive narrative review tying to link neuropsychiatric and inflammatory processes in osteoarthritis. Due to the length of you paper I will only recommend some general issues:
Please ensure figures are cited correctly if you did not produce yourself.
Response: We would like to thank the reviewer for taking his/her time and giving useful comments on our research. We also find it pleasing to see the interest in the topic. Concerning figures, every figure has been produced by the authors of the paper and requires no citations.
The 2nd comment: L.175 Rewording recommended for: "Are also shown to suffer"
Response: We thank the reviewer for his/her suggestion. We have made the rewording of the phrase to “…are also known to experience…” Line 203 in the revised manuscript.
The 3rd comment: If I was able to follow your explanations correctly and based on the fact, that you cited hundreds of papers but some of the essential assumptions are more connected by theory, I would recommend to change in the beginning of your discussion to a less absolute conclusion, like "might play an important role".
Good luck with the paper.
Response: We thank the reviewer for the valuable comment. We do agree that a less absolute and more open-to-discussion conclusion would be a better start for the “Discussion” section. Therefore, we have rephrased the beginning to: “Pertinent literature suggests that low-grade inflammation might play a central role...” Line 629 in the revised manuscript.
Round 2
Reviewer 1 Report
The authors have made the required changes to the manuscript and further have given an explanation of the vast number of citations.
I would still like to see a better version of Fig 4.
Overall, this review is a summary of all the studies on osteoarthritis.